# Niobium Mineralogy of Pliocene A$_1$-Type Granite of the Carpathian Back-Arc Basin, Central Europe

**Monika Huraiová [1], Patrik Konečný [2] and Vratislav Hurai [3],\***



[1]  Department of Mineralogy and Petrology, Comenius University, Faculty of Natural Sciences, Mlynská dolina, Ilkovičova 6, 842 15 Bratislava 4, Slovakia
[2]  State Geological Institute of D. Štúr, Mlynská dolina, 817 04 Bratislava, Slovakia
[3]  Institute of Earth Sciences, Slovak Academy of Sciences, Dúbravská cesta 9, 840 05 Bratislava, Slovakia
\*  Correspondence: vratislav.hurai@savba.sk; Tel.: +00421-2/32293209

**Abstract:** A$_1$-type granite xenoliths occur in alkali basalts erupted during Pliocene–Pleistocene continental rifting of Carpathian back-arc basin (Central Europe). The Pliocene (5.2 Ma) peraluminous calc-alkalic granite contains unusually high concentrations of critical metals bound in Nb, Ta, REE, U, Th-oxides typical for silica-undersaturated alkalic granites, and syenites: columbite-Mn, fergusonite-Y, oxycalciopyrochlore, Nb-rutile, and Ca-niobate (fersmite or viggezite). In contrast, it does not contain allanite and monazite—the main REE-carriers in calc-alkalic granites. The crystallization of REE-bearing Nb-oxides instead of OH-silicates and phosphates was probably caused by strong water deficiency and low phosphorus content in the parental magma. Increased Nb and Ta concentrations have been inherited from the mafic parental magma derived from the metasomatized mantle. The strong Al- and Ca-enrichment probably reflects the specific composition of the mantle wedge modified by fluids, alkalic, and carbonatitic melts liberated from the subducted slab of oceanic crust prior to the Pliocene-Pleistocene rifting.

**Keywords:** alkali basalt; columbite; granite; fergusonite; Pannonian Basin; pyrochlore; xenolith

## 1. Introduction

Niobium belongs to the group of incompatible high-field-strength elements (HFSE) concentrated in carbonatites and peralkalic granites, where it is commonly accompanied by tantalum, yttrium, and rare earth elements (REE). Niobium is classified as a critical high-tech metal, as approximately 90% of the World production is currently exploited from just one mine in Brasil [1].

A-type granites and genetically associated metasomatic rocks and pegmatites are potential resources of critical high-tech metals [2,3]. The original definition of A-type granites [4] included only REE- (excluding Eu), mildly alkalic metaluminous granitoids with low CaO, high FeO$_{tot}$/MgO, prevalence of K$_2$O over Na$_2$O, increased contents of rare-earth (excluding Eu) and other incompatible elements (Nb, Ta, Zr, Hf), and low concentrations of compatible elements (Ba, Sr, Ti, P) originated by the fractional crystallization of alkali basalts at a low oxygen fugacity in intra-plate tectonic settings. The enrichment in HFSE became the essential classification criterion after the definition of within-plate granites [5] and the discrimination of the A-type from other granite types according to the Ga/Al ratio [6]. The A-type clan was consequently subdivided into the A$_1$ granites of continental rifts originated by the differentiation of basalts and the A$_2$ granites of island arcs and continental margins produced by the crustal anatectic melting [7,8]. Correspondingly, the A$_1$-subgroup is diagnostic of the trace element signature similar to that of ocean-island basalts (OIB) and the offset from the OIB field towards the island-arc basalt-derived magmas (IAB) typical of the A$_2$-subgroup reflects the variable degrees of crustal contamination.

Numerous case studies documented that the fractional crystallization of basalt produces ferroan, metaluminous peralkalic, alkalic to alkali–calcic granites with the OIB-like trace element characteristics of the $A_1$-subgroup [9,10] In contrast, $A_2$ granites involve a greater diversity of metaluminous to peralkalic, alkalic to calc-alkalic and ferroan to magnesian compositions [10–13]. The $A_2$-type granite signatures within extensional settings [14] result from the complete or partial melting of the former refractory lower continental crust metasomatized by fluids degassed from the asthenospheric mantle [15].

Peraluminous, calcic to calc-alkalic, ferroan to magnesian A-type granite xenoliths have been recognized in alkali basalts of the Pannonian Basin [16,17]. Despite their great compositional diversity, the granite and associated syenite xenoliths fall within the subgroup of $A_1$-type granites in the discrimination diagrams involving Zr, Nb, Ce, Rb, and Y. The calc-alkalic granites are unique in having increased Nb + Ta contents and locally abundant Nb-rich minerals. As such, the may be considered as fragments of hidden, unconventional resource of critical metals resting in the lithosphere subjacent to the Pannonian Basin. Here we provide the identification of the Nb-bearing phases and classification based on the discriminant analysis. We furthermore define elemental substitutions and discuss possible sources of the critical metals.

## 2. Geological Setting

The Pannonian Basin is the intra-Carpathian back-arc basin originated after the Middle Miocene subduction compensated by the diapiric uplift of asthenosphere [18,19]. Rifting in the orogenic hinterland and concomitant partial melting of the uplifted asthenospheric and lithospheric mantle resulted in a within-plate, post-collisional Na-alkali basalt volcanism concentrated mostly in the western and northern part of the Pannonian Basin, where magmatic and phreatomagmatic eruptions from isolated monogenetic or short-lived volcanic centers have created numerous maars, cinder cones, and lava flows dated from 7 to 0.2 Ma [20,21].

The alkali basalt volcanism in the northern part of the Pannonian Basin (Figure 1) hosts numerous volcanic centers erupted within the time interval from Upper Miocene (Pontian) to Pleistocene [22]. Volcanic eruptions in this area ejected fragments of peridotites, mafic cumulates, syenites and granites that carry invaluable information about the composition of lithospheric mantle and crust affected by partial melting, metasomatism, basaltic magma underplating and differentiation [16,17,23–25]. The northern part of Pannonian Basin is recently characterized by an increased heat flow (80–90 $\text{mWm}^{-2}$) within the thinned crust (26–28 km) underlain in the depth of ~70 km by asthenospheric bulge [26–28].

Rounded or angular granite fragments, up to 20 cm in diameter (Figure 2), with increased contents of critical metals, have been found in the frontal part of the alkali basalt lava flow discharged from the Monica volcanic center (589 m, a.s.l. (above sea level)). The basalt flow is exposed in the stone pit (N 48°14′9.99″, E 19°51′43.18″, 436 m a.s.l.) near the Čamovce village. Age of the lava flow determined by the K/Ar whole-rock method overlaps the interval of 4.3–6.1 Ma, including uncertainty limits [29].

Trace element abundances, zircon U–Pb–Hf data, electron probe microanalyses of major rock-forming and selected accessory minerals (Fe–Ti oxides, zircon, thorite, and pyrochlore) in the granite xenoliths from Čamovce have been published elsewhere [16,17]. Granite xenoliths are composed of fine-grained anhedral quartz, alkali feldspar spanning the compositional range from oligoclase ($Ab_{80}Or_{13}An_7$) to sanidine ($Or_{51}Ab_{48}$), and clusters of interstitial glass. Magnetite phenocrysts, several mm in size, are macroscopically discernible in hand specimens and thin sections. Apart from diagnostic trace element abundances plotting within the field of $A_1$-type granites with OIB-like trace element signatures [16], the genetic link of granite xenoliths with the Pliocene intra-plate magmatism and mantle-derived mafic magmas has also been demonstrated by the 5.18 ± 0.02 Ma old zircon with strongly positive εHf values of 14.2 ± 3.9 [17].

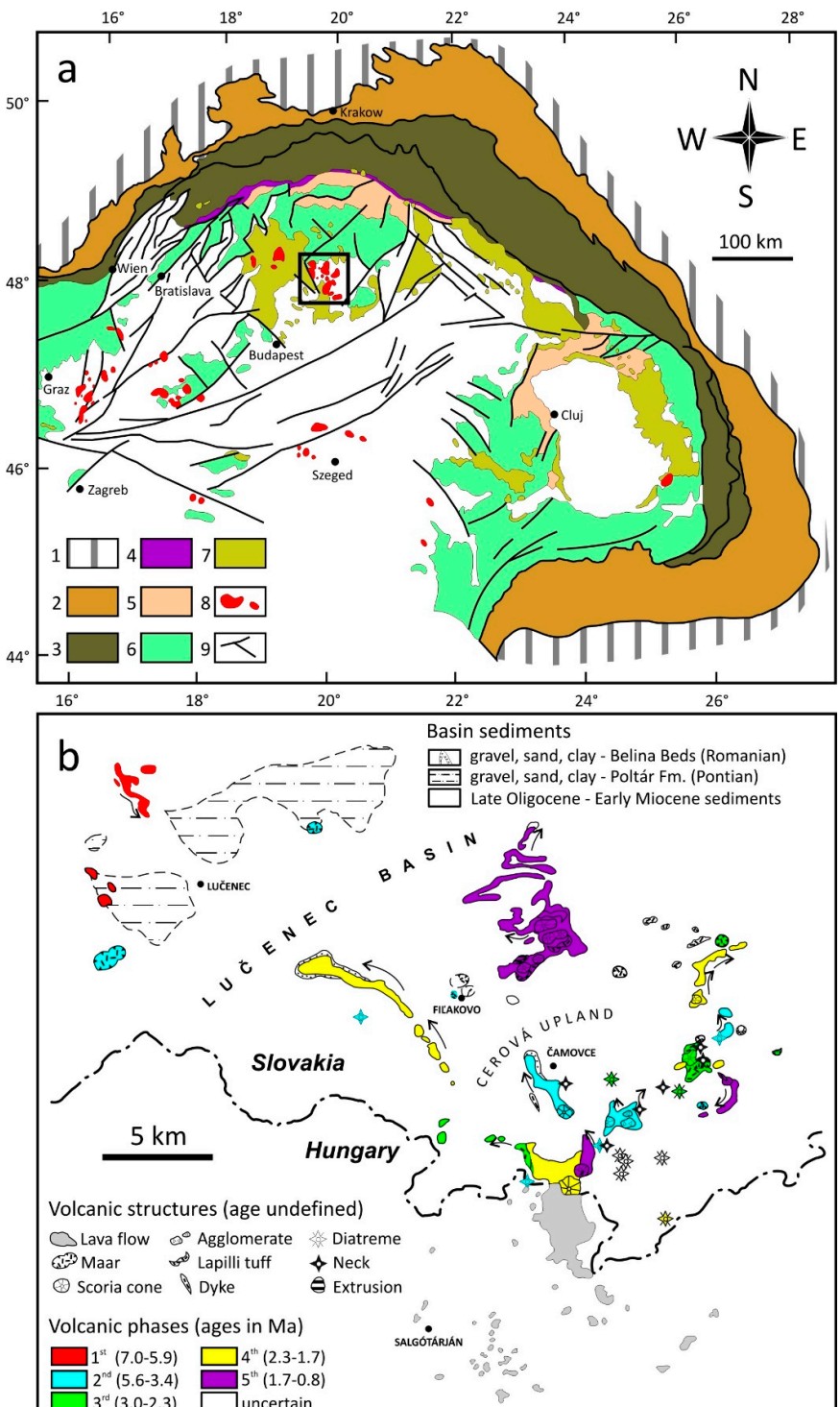

**Figure 1.** (**a**) Schematic drawing of the Carpathian arc and the intra-Carpathian back-arc basin modified after [22]: 1: European platform; 2: foredeep; 3: Tertiary accretionary wedge; 4: Klippen belt, 5: Palaeogene back-arc basin; 6: Variscan crystalline basement with pre-Tertiary sedimentary cover and Mesozoic nappe units (Cretaceous accretionary wedge); 7: Neogene andesites and rhyolites; 8: Upper Miocene-Quaternary alkali basalt volcanism; 9: major normal, strike-slip and thrust faults. The rectangle marks the South Slovakian–North Hungarian Volcanic Field. (**b**) Enlarged view of the study area with volcanic phases as indicated by K/Ar dating of lava flows [30], U/Pb zircon, U/Th monazite, (U-Th)/He zircon and apatite dating of maars ([21,31–33] and unpublished data).

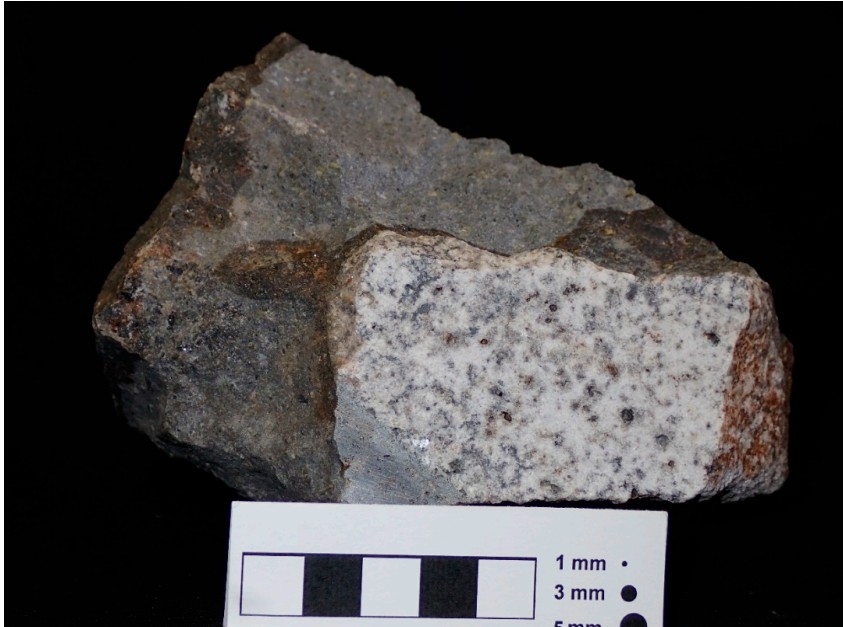

**Figure 2.** Granite xenolith in basalt from the Čamovce quarry.

## 3. Methods

Chemical compositions of Nb-bearing minerals was determined using the electron probe micro-analysis (EPMA). The EPMA of Nb-rich minerals is a challenging task due to multiple interferences within the group of high-field-strength elements. A CAMECA SX-100 electron probe micro-analyser in wavelength-dispersive (WDS) mode was employed. An accelerating voltage of 15 kV was used to optimize the spatial resolution, matrix correction factors, and to minimize the sample surface damage. Beam diameters ranged between 2 μm and 5 μm. Excitation lines, crystals, calibrants, average detection limits and average standard deviations are listed in the electronic Supplementary file (Table S1). Matrix effects were resolved using the X-PHI correction method [34]. Spectral lines unaffected by interferences were selected, however, when peak overlaps existed, empirical correction factors [35,36] were applied.

## 4. Results

### 4.1. Characterization of Nb-Bearing Phases

HFSE-bearing minerals have been recognized microscopically, using polarizing microscope and electron probe imaging (Figure 3). Anhedral to euhedral magnetite is either homogeneous or consists of porous cores and compact overgrowths. Other Fe–Ti(±Nb,Ta) oxides occur as relics of early ilmenite replaced by rutile, rutile intergrowths with the early ilmenite, rutile exsolutions within early ilmenite, rutile-ilmenite-silicate symplectite around resorbed ilmenite grains, and compact rutile rims around columbite associated with the early ilmenite. Late ilmenite occurs together with rutile, orthopyroxene, quartz, and sanidine in the interstitial silicate glass. Ilmenite is an early-crystallizing phase, but its temporal relationship with magnetite and zircon remains unclear.

Zircon crystals, up to 500 μm in size, are coeval with late Ti-magnetite. The zircon grains consist of opaque cores overcrowded with uranothorite inclusions. BSE images (Figure 3d) document sector-zoned distribution of uranothorite inclusions, indicating a breakdown of the $ZrSiO_4$–$ThSiO_4$ solid solution during the initial zircon growth. The opaque uranothorite-rich core is overgrown by a clear inclusion-free rim. Weak magmatic oscillatory growth zoning is discernible in BSE images, whereas cathodoluminescence imaging did not reveal any zoning.

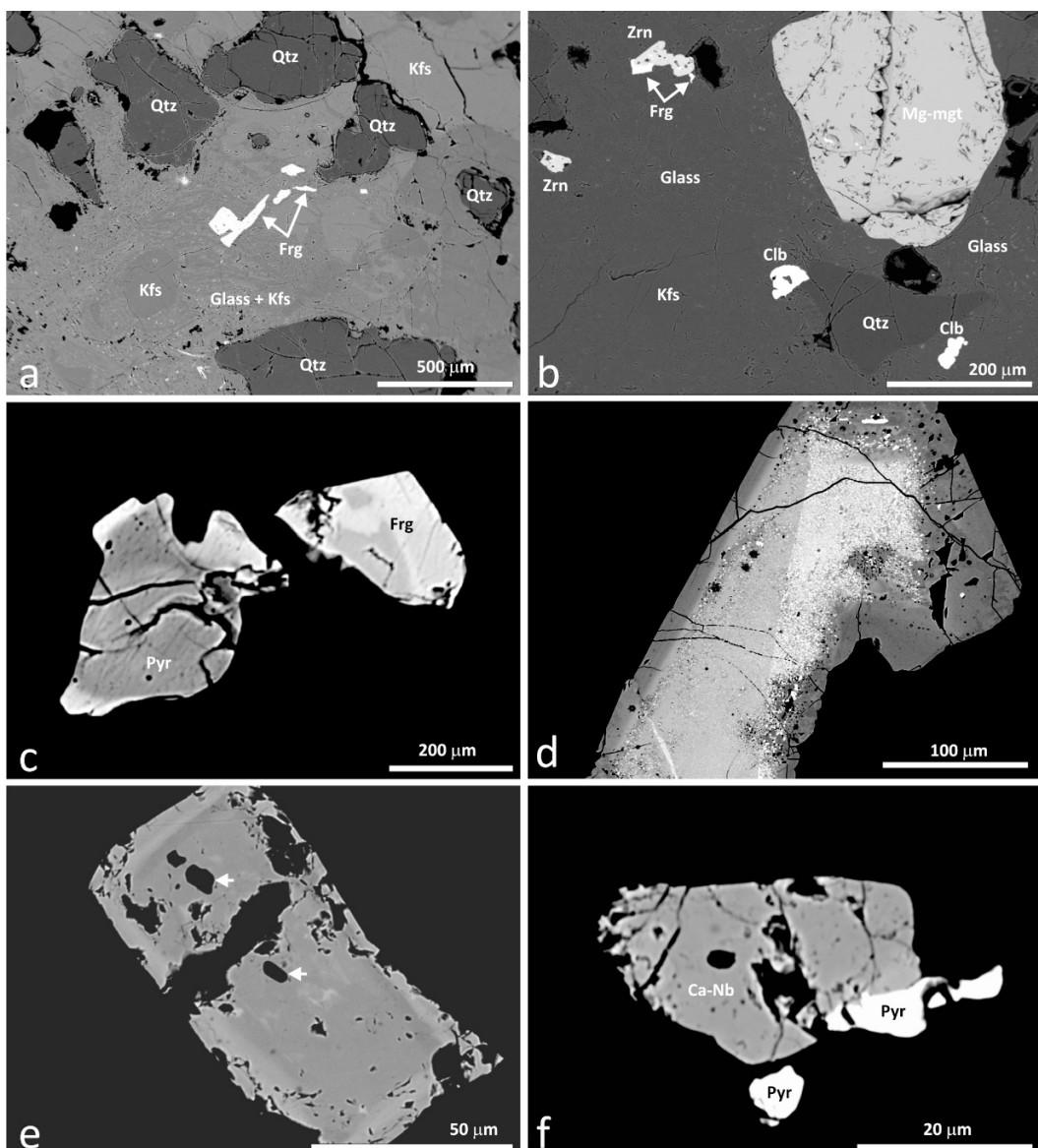

**Figure 3.** Back-scattered electron (BSE) images of high-field-strength elements (HFSE)-bearing minerals in granite xenoliths. (**a**) Fergusonite (Frg) cluster in the interstitial glass. Qtz and Kfs denote quartz and potassium feldspar, respectively. Sample Ca10. (**b**) Magnetite phenocryst with increased Mg content (Mg-mgt) associated with zircon (Zrn) and columbite (Clb). Fergusonite (Frg) sticks to zircon grains with uranothorite inclusions. Sample Ca3. (**c**) Complex domain zoning of pyrochlore and fergusonite caused by variable Ti, Nb and U contents. Sample Ca3. (**d**) Sector-zoned zircon with an inclusion-free rim around the core overcrowded with tiny uranothorite inclusions. Isolated bright spots in the rim are pyrochlore inclusions. (**e**) Oscillatory growth zoning of fergusonite crystal with two silicate glass inclusions indicated by white arrows. Sample Ca3. (**f**) Ca-niobate (Ca–Nb) associated with pyrochlore (Pyr). Sample Ca3c.

(Y,REE,U,Th)–(Nb,Ta,Ti)-oxides are opaque to brown and translucent along margins in transmitted light. They form isolated anhedral to subhedral isometric grains and clusters enclosed in the interstitial glass and rock-forming quartz and feldspars. Oscillatory growth and domain zonings are discernible in BSE images (Figure 3c,e). (Y,REE,U,Th)–(Nb,Ta,Ti)-oxides post-date the Th-rich zircon cores, but they are coeval with clear zircon rims. Crystallization succession scheme of Fe–Ti(±Nb,Ta) and (Y,REE,U,Th)–(Nb,Ta,Ti)-oxides based on observations of spatial relationships in thin sections is summarized in Figure 4.

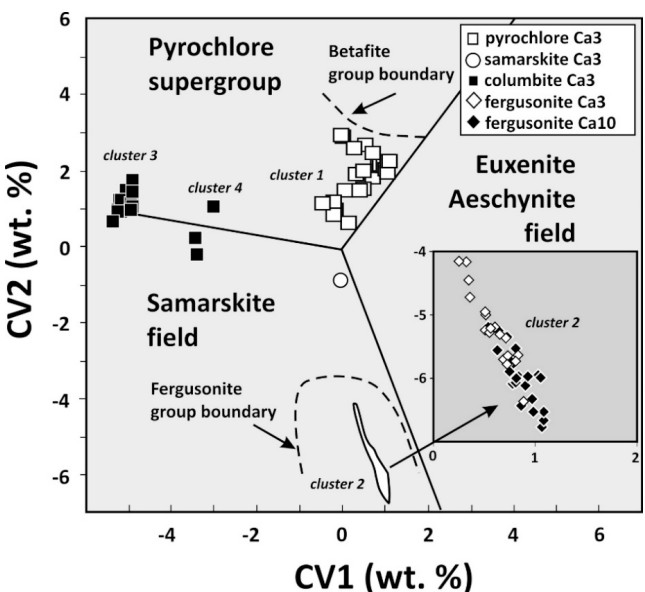

**Figure 4.** Crystallization succession scheme of HFSE-rich minerals and Fe–Ti oxides.

Analysis of EPMA data (Electronic Supplementary file, Tables S2–S4) revealed a total of four clusters and one outlier (Figure 5). Table S1 refers to cluster 1, projecting within the pyrochlore super-group field. Hence, crystallochemical formulas were recalculated from EPMA as $AB_2X_6Y$, such that large [8]-coordinated cations (Na, Ca, Sr, Ba, $Fe^{2+}$, Pb, Y, REE, U, Th, ± vacancy, ± $H_2O$), were affiliated to the *A*-site, [6]-coordinated, high-field-strength cations (Nb, Ta, Ti, W, Zr, Hf, $Fe^{3+}$, Mg, Al, and Si) without vacancies were attributed to the *B*-site, *X*-site contained O plus subordinate OH and F, and *Y*-site O, OH, $H_2O$, F, vacancy, and very large cations, such as K, Cs, Rb [37]. The dominant species in the *Y*-site defined a primary prefix of the mineral name. The secondary prefix was derived from the dominant *B*-site cation. Structural formulas of pyrochlore listed in Table S2 are charge-balanced, with the fixed number of 2 cations in the vacancy-free *B*-site. The maximum OH concentration that could be accommodated in the crystal structure was calculated as an equivalent to the sum of deficits in the *A*- and *Y*-sites. The calculated OH content in apfu was then recalculated to wt. % $H_2O$. Fluorine concentrations were always below the detection limit of 0.03 wt. %.

**Figure 5.** Discrimination plot for pyrochlore-, samarskite-, fergusonite- and euxenite–aeschynite group minerals [38] with projections of EPMA data from Čamovce. CV1 = 0.245$Na_2O$ + 0.106CaO – 0.077*Fe*\* + 0.425PbO + 0.22$Y_2O_3$ + 0.28LREE$_2O_3$ + 0.137HREE$_2O_3$ + 0.1*U*\* + 0.304TiO$_2$ + 0.097Nb$_2O_5$ + 0.109*Ta*\* – 12.81; CV2 = 0.102$Na_2O$ – 0.113CaO – 0.371*Fe*\* – 0.167PbO – 0.395$Y_2O_3$ – 0.28LREE$_2O_3$ – 0.265HREE$_2O_3$ – 0.182*U*\* – 0.085TiO$_2$ – 0.166Nb$_2O_5$ – 0.146*Ta*\* + 17.29 (oxide wt. %), where *Fe*\* = FeO + Fe$_2O_3$ + MnO, *U*\* = UO$_2$ + ThO$_2$, *Ta*\* = Ta$_2O_5$ + WO$_3$ (wt. %). Provisional boundary of the betafite group is based on published EPMA data on oxycalciobetafite [39–41].

Maximum OH$_{(A + Y)}$ concentrations have been sometimes greater than 0.5 apfu, however, owing to the deficient *A*-site, only one analysis (Ca3d, an1) showed the predominance of OH in the *Y*-site,

assuming no vacancies. Due to negligible F and extremely large cations (K, Rb, Cs), oxygen must have been the dominant anion at the *Y*-site in all but one analyses. Since calcium (0.774–1.013 apfu) predominated in the *A*-site, analyzed minerals could be named as oxycalciopyrochlore according to the dominant constituent rule [42] and the criteria summarized in [37]. The remaining analysis (Ca3d, an1) corresponds to hydroxycalciopyrochlore.

One outlier within the samarskite field (open circle in Figure 5) shows enrichment in Fe, Mn, Th, Y, and depletion in Na compared to the cluster 1. The calcium predominance in the *A*-site over Y + REE and U + Th is diagnostic of calciosamarskite [43], however, structural formula calculated with four oxygen atoms (*AB*O$_4$) showed strong excess of cations summing at 2.28. On the other hand, the cation-to-oxygen proportion was similar to that of pyrochlore, though the empirical formula recalculated using 7 oxygen atoms and all iron as FeO was still deficient in the *B*-site with the fully occupied *A*-site. Hence, ferric iron was provisionally calculated assuming the full occupancy of *X*- and *Y*-sites by oxygen and full occupancies of the *A*- and *B*-sites (2 apfu). The final crystallochemical formula corresponds to the almost ideal $A_2B_2O_7$ pyrochlore stoichiometry devoid of vacancies and hydroxyl groups.

EPMA data from Table S2 project within the fergusonite subfield in Figure 5 marked as the cluster 2. Weakly oscillatory growth-zoned fergusonite rarely contained homogeneous silicate glass inclusions (Table 1), corresponding to a peraluminous, subalkalic (TAS scheme), calcic and ferroan rhyolite.

**Table 1.** Chemical compositions of interstitial glass, fergusonite-hosted glass inclusion (MI) and the xenolith host. Sample Ca3, data from [17].

| Composition (wt. %) | Bulk | Interstitial | Interstitial | Interstitial | MI in frg |
|---|---|---|---|---|---|
| SiO$_2$ | 75.54 | 68.82 | 70.52 | 69.21 | 73.17 |
| TiO$_2$ | 0.04 | 0.01 | 0.01 | bdl | 0.03 |
| Al$_2$O$_3$ | 14.07 | 15.68 | 15.21 | 17.26 | 12.66 |
| FeO | 0.69 | 0.09 | 0.15 | 0.17 | 0.40 |
| MnO | 0.03 | 0.00 | 0.08 | 0.01 | 0.16 |
| MgO | 0.03 | 0.02 | 0.01 | bdl | bdl |
| CaO | 0.78 | 0.21 | 0.10 | 0.19 | 0.60 |
| Na$_2$O | 4.64 | 2.95 | 3.97 | 4.64 | 2.79 |
| K$_2$O | 3.44 | 7.32 | 6.94 | 8.39 | 3.22 |
| Cl | na | 0.04 | 0.04 | 0.02 | na |
| O=Cl | | −0.01 | −0.01 | −0.01 | |
| Total | 99.26 | 95.13 | 97.02 | 99.88 | 93.03 |
| **Normative minerals** | | | | | |
| **(wt. %)** | | | | | |
| *quartz* | 33.34 | 24.50 | 21.30 | 9.67 | 46.10 |
| *albite* | 39.55 | 25.94 | 34.33 | 39.16 | 25.38 |
| *anorthite* | 3.9 | 1.10 | 0.51 | 0.94 | 3.20 |
| *K-feldspar* | 20.48 | 45.49 | 42.28 | 49.65 | 20.45 |
| *corundum* | 1.3 | 2.71 | 1.07 | 0.23 | 3.76 |
| *hypersthene* | 1.34 | 0.21 | 0.45 | 0.33 | 1.06 |
| **Total-alkali-silica (TAS)** | | | | | |
| **Scheme [44]** | | | | | |
| *A/NKC* | 1.10 | 1.19 | 1.07 | 1.01 | 1.38 |
| *NK/A* | 0.81 | 0.81 | 0.92 | 0.97 | 0.64 |
| prefix 1 | potassic | potassic | potassic | potassic | potassic |
| prefix 2 | peraluminous | peraluminous | peraluminous | peraluminous | peraluminous |
| prefix 3 | subalkalic | alkalic | alkalic | alkalic | subalkalic |
| TAS name | rhyolite | rhyolite | rhyolite | rhyolite | rhyolite |
| **Frost et al. Scheme [45]** | | | | | |
| *Fe* * | 0.96 | 0.82 | 0.96 | 1.00 | 1.00 |
| *MALI* | 7.35 | 10.57 | 11.14 | 12.85 | 5.82 |
| prefix 4 | ferroan | magnesian | ferroan | ferroan | ferroan |
| prefix 5 | calc-alkalic | alkalic | alkalic | alkalic | calcic |

na: not analyzed; bdl: below detection limit; *A/NKC* = Al$_2$O$_3$/(CaO+Na$_2$O+K$_2$O-1.67P$_2$O$_5$); *NK/A* = (Na$_2$O + K$_2$O)/Al$_2$O$_3$; *Fe* * = (FeO + MnO)/(FeO + MnO + MgO); *MALI* = (Na$_2$O + K$_2$O)/CaO.

The ideal formula and the site occupancy of fergusonite are identical with that of pyrochlore, however, vacancies are not present. Indeed, EPMA data recalculated with four oxygen atoms (*AB*O$_4$)

resulted in the almost full cation occupancy of the *A*- and *B*-sites. The predominance of yttrium over REE's is diagnostic of fergusonite-Y.

Clusters 3 and 4 in Figure 5 belong to the columbite family. The Fe,Mn-rich variety of cluster 3 is the most abundant Nb-rich mineral besides fergusonite and pyrochlore. Isolated anhedral to subhedral grains visualized in BSE show domain zoning caused by fluctuating Nb/Ta ratios (Figure 2). EPMA data and structural formulas (Table S3) recalculated to 3 cations and 6 oxygen atoms ($AB_2O_6$) are consistent with columbite. Prevalence of Mn over $Fe^{2+}$ in the *A*-site is diagnostic of columbite-Mn. Mn# = Mn/(Mn + $Fe^{2+}$) and Ta# = Ta/(Ta + Nb) ratios cluster at 0.56–0.71 and 0.01–0.14, respectively, without any fractionation trend.

Calcium-dominated niobate (Table S3, cluster 4 in Figure 5) is intimately intergrown with columbite-Mn or pyrochlore. Apart from an increased Ca content, it is also typical by enrichment in Mn (up to 4.5 wt. % MnO, 0.21 apfu, Mn# 0.76–0.84) compared to columbite. Simplified formula corresponding to $(Ca,Mn)Nb_2O_6$ refers to fersmite or vigezzite. Fersmite and vigezzite are dimorphic, albeit both with orthorhombic symmetry. While the $P_{mnb}$ space group vigezzite [46] belongs to the aeschynite group, the $P_{can}$ space group fersmite is a member of the euxenite group [47]. However, EPMA data from granite xenoliths project on the boundary between pyrochlore and samarskite groups.

The remarkably high Mn content in the Ca–niobate from granite xenoliths contrasts with the Mn-poor fersmite/viggezite described in carbonatites [48] and fractionated albite-rich granite pegmatites [46,49–53]. An intimate intergrowth of fersmite/viggezite with columbite might indicate that the Ca–niobate from Čamovce is simply a Ca-analogue of columbite [54]. Typical for the Ca–niobate from Čamovce is also the depletion in REE, particularly Ce, similar to the low-REE fersmite/vigezzite from a lithium pegmatite [55] and carbonatite [48].

Rutile is the least abundant Nb-concentrating mineral in the granite xenoliths. The highest Nb and Ta concentrations (up to 51 wt. % oxides total, 0.37 apfu) were recorded in the rutile Rt3 crystallized along ilmenite-columbite contacts. Increased Nb concentrations were also identified at contacts of rutile exsolutions (Rt1) with ilmenite host, and locally also in the non-stoichiometric rutile-resembling phases (Rt2) replacing the early ilmenite. Compositions of the non-stoichiometric phases project between ilmenite, pseudobrookite, pseudorutile and ilmenorutile endmembers (Figure 6). Newly formed rutile Rt4 crystallizing together with Ilm3 within the interstitial glass is almost Nb-free.

A small amount of Nb is also bound in early ilmenite, which contains significant Mn (3.5–11.8 wt. % MnO, up to 25 mol. % pyrophanite endmember $MnTiO_3$), substituting for Fe in the *A*-site. The *B*-site electroneutrality with up to 3.9 wt. % $Nb_2O_5 + Ta_2O_5$ (0.01–0.04 apfu) is counterbalanced by the trivalent iron (up to 4.5 wt. % $Fe_2O_3$). In contrast, late interstitial ilmenite is practically devoid of Nb.

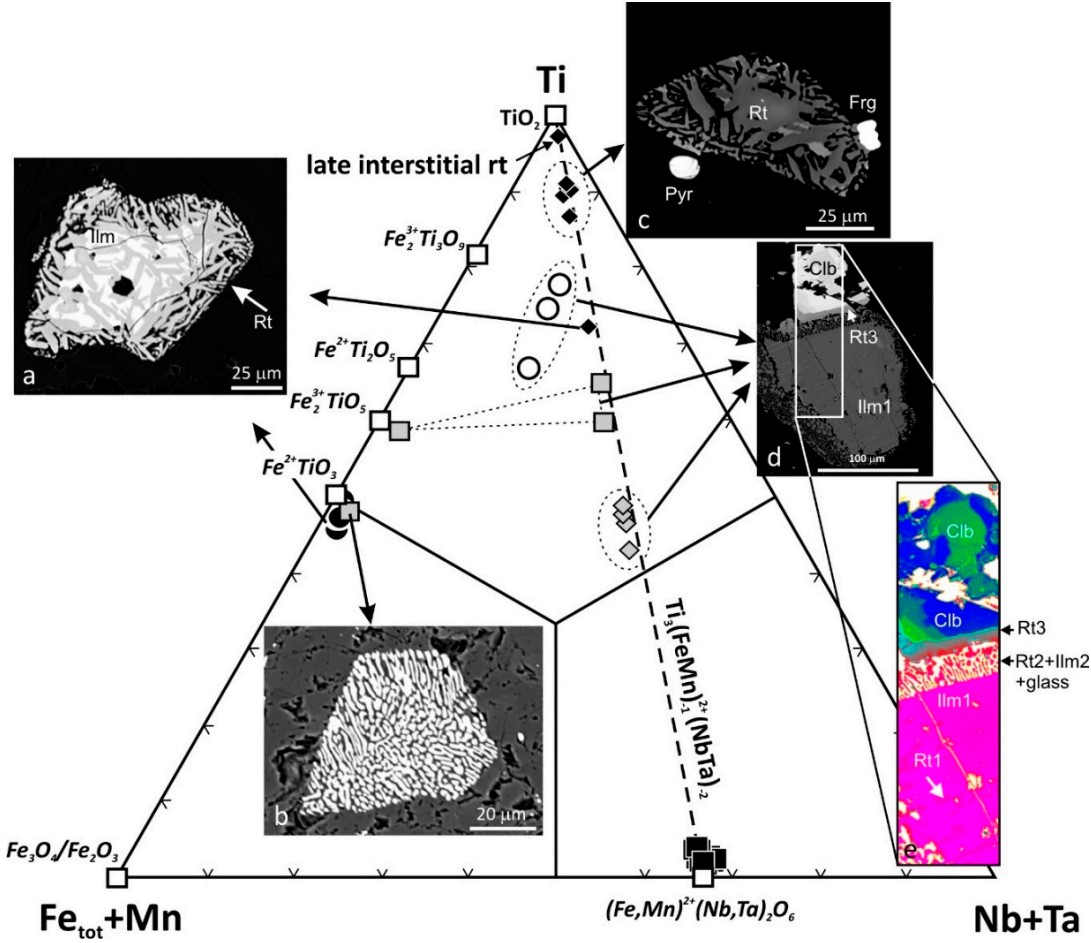

**Figure 6.** The triangular plot of Ti–(Fe,Mn)–(Nb,Ta) atomic proportions with projection points of ideal endmembers (open squares, formulas in italics) and EPMA data on Nb,Ti,Fe,Mn-oxides from Čamovce [17]. Open circles represent rutile exsolutions (Rt1) in ilmenite (Ilm1), shaded squares are symplectic ilmenite (Ilm2) and rutile (Rt2) replacing the early ilmenite Ilm1, shaded diamonds are compact rutile rims (Rt3) between ilmenite and columbite, solid diamonds correspond to the euhedral rutile replacing early ilmenite (solid circles). Selected rutile and ilmenite types are documented in inserted BSE images (**a**–**d**). The false color-coded sector (**e**) visualizes the rutile-ilmenite-silicate glass symplectite grading into the compact rutile, as well as Nb–rutile exsolutions within the ilmenite host. Niobium and tantalum contents tend to increase from Ilm1 (up to 3.8 wt. % $Nb_2O_5$ + $Ta_2O_5$), through Rt1 (up to 22 wt. %), Rt2 (up to 37 wt. %), Rt3 (47–51 wt. %), to columbite (78 wt. %). Blue domains in the columbite are enriched in Ta (up to 16.5 wt. % $Ta_2O_5$). Ilm1 is enriched in Mn (up to 11.8 wt. % MnO) and Sm (1.1 wt. % $Sm_2O_3$).

### 4.2. Crystal Chemistry of Nb-Bearing Phases

Strong correlations at the rutile-columbite join have been recorded for Nb–Ta, Nb–Ti, Nb–Y and Mn–$Fe_{tot}$ couples. The linear trend from rutile to columbite endmembers (Figure 6) indicates a heterovalent substitution of three Ti atoms for one divalent and two pentavalent cations [56], combined with two homovalent $Mn^{2+}$–$Fe^{2+}$ and $Ta^{5+}$–$Nb^{5+}$ substitutions. The correlation coefficient for the exchange vector $Ti_3(Nb,Ta)_{-2}(Fe^{2+},Mn)_{-1}$ equals to $r^2 = 0.63$, assuming all Fe as $Fe^{2+}$. A moderate displacement from the ideal substitution line can be explained by the subordinate substitution expressed by the exchange vector $Ti_2(Y^{3+})_{-1}(Nb,Ta)_{-1}$ ($r^2 = 0.53$). Correlation has not been found for another possible exchange vector $Ti_2(Fe^{3+})_{-1}(Nb,Ta)_{-1}$ ($r^2 = 0.04$).

Substitutions in the fergusonite-group seem to be more complicated than those along the rutile-columbite join. Two-element correlation coefficients (Table 2) reveal two distinct groups of

rare earth elements, LREE (La–Eu) and HREE (Gd–Lu) series, with an antithetical behavior against Ca, Ti and Th. The similar contrasting behavior of MREE against LREE and HREE has already been observed in aeschynite [57]. Yttrium behaves in accord with the HREE group due to similar crystal ionic radii of Y and Gd–Lu, 104 pm and 107–100 pm respectively, compared to much larger ionic radii of the La–Eu group (117–109 pm).

**Table 2.** Correlation coefficients ($r^2$) for major elements in fergusonite.

|        | Ca    | Ti    | Th    | Nb    |
|--------|-------|-------|-------|-------|
| W      | −0.50 | −0.39 | −0.57 | 0.18  |
| Nb     | −0.25 | −0.62 | −0.28 | 1.00  |
| Ta     | 0.02  | 0.10  | 0.03  | −0.25 |
| Ti     | 0.52  | 1.00  | 0.54  | **−0.62** |
| Th     | **0.90** | 0.59  | 1.00  | −0.28 |
| U      | 0.00  | 0.01  | 0.00  | −0.13 |
| Y      | **−0.87** | **−0.66** | **−0.96** | 0.27  |
| Ce     | 0.55  | 0.50  | 0.72  | −0.16 |
| Pr     | 0.53  | 0.41  | 0.68  | −0.11 |
| Nd     | 0.36  | 0.33  | 0.54  | −0.07 |
| Sm     | 0.04  | 0.02  | 0.11  | −0.01 |
| Gd     | −0.18 | −0.07 | −0.07 | 0.05  |
| Tb     | −0.30 | −0.15 | −0.23 | 0.08  |
| Dy     | −0.60 | −0.29 | −0.42 | 0.26  |
| Ho     | −0.41 | −0.15 | −0.29 | 0.22  |
| Er     | −0.88 | −0.47 | −0.81 | 0.35  |
| Tm     | −0.61 | −0.42 | −0.59 | 0.21  |
| Yb     | −0.53 | −0.31 | −0.62 | 0.11  |
| Lu     | −0.25 | −0.32 | −0.38 | 0.15  |
| Mn     | 0.28  | 0.04  | 0.45  | −0.06 |
| Ca     | 1.00  | 0.37  | **0.90** | −0.25 |
| LREE   | 0.39  | 0.27  | 0.59  | −0.03 |
| HREE   | **−0.92** | **−0.46** | **−0.78** | 0.34  |
| Y + HREE | **−0.94** | **−0.62** | **−0.97** | 0.32  |

The highest correlation coefficient ($r^2 = 0.97$) pertains to the charge-balanced substitution $Ca^{2+}{}_A$ + $Th^{4+}{}_A \Leftrightarrow 2(Y+HREE)^{3+}{}_A$, occurring at the *A*-site only. The coupled substitution $Ca^{2+}{}_A$ + $Nb^{5+}{}_B$ $\Leftrightarrow REE^{3+}{}_A$ + $Ti^{4+}{}_B$ (REE = Pr, Nd, Sm) proposed for the aeschynite-group [57] may play a role also in our samples with REE = Y + (Gd–Lu) ($r^2 = 0.7$). Other coupled heterovalent substitutions also exhibit high correlation coefficients > 0.95, e.g., $Ca^{2+}{}_A$ + $Ti^{4+}{}_B \Leftrightarrow 2REE^{3+}{}_A$, $Th^{4+}{}_A$ + $Ti^{4+}{}_B \Leftrightarrow 2REE^{3+}{}_A$ + $Ca^{2+}{}_A$. In contrast to HREE, LREE's are poorly correlated with other elements. The highest correlation coefficient $r^2 = 0.41$ pertains to the charge-balanced exchange vector $Ca^{2+}$ $Th^{4+}$ $(LREE)^{3+}{}_{-2}$.

Unlike columbite and fergusonite, major elements in calciopyrochlore exhibit poor correlations and also the non-systematic behavior of rare elements, most likely owing to low concentrations affected by increased analytical uncertainty. Together with the homovalent Nb for Ta substitution ($r^2 = 0.61$), increased correlation also exists for the heterovalent $(Fe + Mn)^{2+}$ for $2Na^+$ substitution ($r^2 = 0.66$).

## 5. Discussion

Granite xenoliths from Čamovce correspond to the peraluminous calc-alkali granite, except for one barren xenolith devoid of HFSE-bearing phases (except zircon), which paradoxically projected within the alkali granite field [17]. The assemblage of HFSE-bearing minerals dominated by pyrochlore, fergusonite, samarskite, and columbite is thus typical of alkalic granites [58,59]. Allanite as the main Y-REE carrier in calc-alkali peraluminous granites [60] is missing in the granites from Čamovce, most likely due to a strong water deficiency (<2 wt.%) in the parental magma needed for the

crystallization of OH-bearing silicates. Monazite is also absent because of strong depletion in phosphorus (<0.02 wt. % $P_2O_5$) removed probably by an early apatite fractionation.

Primary magmatic pyrochlore is abundant in kimberlites [61], strongly differentiated leucogranites [62–64], Be-rich granite pegmatites [65], carbonatites and their fenite aureoles [66], nepheline syenites and nephelinites [40,67], and lunar regolith [68]. In spite of the relatively high frequency of pyrochlore-group minerals in magmatic rocks, there are only a few examples of oxycalciopyrochlore [69–73]. Similar mineral, oxycalciobetafite, has been hitherto described only in lunar granophyre [74], miarolitic cavities within a foid-bearing syenite [39,40], and hydrothermal veins intersecting dolomite marbles in the contact aureole of the Adamello Massif, Italy [41].

Pyrochlore from Čamovce granite is specific by an increased Ca content and the oxygen-dominated occupancy of the *Y*-site, reflecting the calcic- to the calc-alkalic composition of the parental melt and the low water activity during crystallization. Untypical is also the depletion in Sr, Zr, and F diagnostic of pyrochlore from carbonatites and genetically related alkali granites, although the analyses from Čamovce granites project close to or within the carbonatite field defined in the Nb–Ti–Ta discrimination diagram [75] discussed in [17].

The presence of Nb-dominant member of the pyrochlore-supergroup instead of the Ta-rich microlite is also untypical concerning the peraluminous composition and the advanced fractionation of the Čamovce granite. The highest modal abundance of Nb–U–REE-bearing accessory minerals has been found in the most differentiated granite xenolith with the highest bulk Rb (121 ppm) and Nb + Ta concentrations (223 ppm). Strongly positive Nb + Ta, Zr + Hf, and U + Th anomalies also indicate well-fractionated granites. Granite xenoliths from Čamovce also show negative Ba, Sr, Eu and Ti anomalies complementary with positive anomalies of these elements in mafic cumulate xenoliths [16,17]. The depletion in LILE and Ti indicates the fractionation of feldspars and (Fe)Ti-oxides, respectively, and is thus consistent with the magmatic differentiation from mafic parental melts. The local enrichment in REE-bearing phases in granite xenoliths from Čamovce results in trace element distribution patterns that do not match those in typical granites. Particularly intriguing is the enrichment in medium and heavy REE's (Gd–Lu) in well-fractionated xenoliths contrasting with the enrichment in light REE and depletion in MREE and HREE in barren xenoliths [16,17]. The variable REE patterns are mainly controlled by the abundance of fergusonite – the main concentrator of medium and heavy rare earth elements.

Samarskite-Y, fergusonite-Y and niobian rutile are also typical accessory minerals in Cretaceous post-collisional $A_2$-type granites of Western Carpathians [76]. Locally abundant pyrochlore-group minerals occur in pegmatites derived from Variscan calc-alkalic S- and I-type orogenic granites of Western Carpathians [77,78]. Accessory pyrochlore was also found in the metasomatic albitite associated with the Permian post-orogenic S-type granite of Western Carpathians [79]. Compared with Čamovce granites, the samarskite and fergusonite from the above-mentioned occurrences are depleted in Th, U and Ca, and enriched in Fe and Y. The principal genetic difference, however, stems in the fact, that samarskite and fergusonite in the Čamovce granites are primary magmatic phases, whereas those in other Carpathian granites represent post-magmatic alteration and metamorphic products, crystallizing at the expense of primary magmatic Nb–rutile and columbite.

Extremely consistent U–Pb LA-ICP-MS ages of zircons recovered from the Čamovce granite, 5.18 ± 0.02 Ma, overlap the period of early Pliocene rifting within the Pannonian Basin and refine the lower limit for the effusion age of the host basalt indicated by K-Ar data. The age-corrected εHf values in zircons, from 8.1 ± 0.5 to 21.1 ± 1.7, with a mean at 14.2 ± 3.9, unequivocally indicate mantle-derived parental melts uncontaminated by the crustal material in spite of the peraluminous, calc-alkalic granite composition reminiscent of crustal anatectic melts [17]. There is a clear tendency of an increasing alkalinity from the essentially calcic composition of glass inclusions in fergusonite, through the bulk calc-alkalic composition of xenoliths, to the dominantly alkalic character of the interstitial glass (Table 1), suggestive of an addition of peralkalic melt or fluid component responsible for the crystallization of HFSE-bearing minerals either in the source region or during the interaction of xenoliths with alkali

basalt. The U–Pb zircon age overlapping with the effusion K-Ar age of the host basalt seems to favor the second possibility. This interpretation is, however, invalidated by the low Nb content in the late rutile and ilmenite crystallized together with quartz, orthopyroxene and probably also sanidine from the interstitial melt. Moreover, in contrast to HFSE-rich accessory phases, the interstitial minerals exhibit skeletal growth indicative of rapid crystallization from an undercooled liquid [17]. Hence, the alkalic character of the interstitial melts most likely results merely from the preferential melting of alkali feldspars during the xenolith transport in the hot basaltic magma, whereas the calcic character of fergusonite-hosted melt inclusions may result from the loss of alkalis during analyses with the focused electron beam.

The fractional crystallization of within-plate basalts terminates with ferroan, metaluminous, peralkalic, alkalic to alkali–calcic granites [9,10]. Owing to the extremely low Mg contents close to analytical detection limits, all Čamovce xenoliths including their interstitial and inclusion melts can be regarded as ferroan, being thus consistent with the differentiation from the ferroan mafic precursor. However, the peraluminous calc-alkalic character does not overlap with mantle-derived alkalic mafic melts.

The occurrence of Nb-rich granite xenoliths in the northern part of the Pannonian Basin overlaps the region with Nb-rich amphiboles detected in metasomatized peridotite xenoliths ejected in similar basalts [80]. From a total of three metasomatic events recognized, the Nb-poor amphiboles are believed to record the earliest metasomatic event coincidental with the infiltration of Miocene subduction-related fluids into the overlying mantle wedge. The superimposed U–Th–(Nb–Ta)–REE enrichment coeval with the extension of the Pannonian Basin gave rise to the crystallization of Nb-rich amphiboles in the metasomatized mantle wedge. It seems thus likely that the fractional crystallization of the underplated Nb-rich alkali basalts mingled with a Ca-rich, mantle-derived contaminant (carbonatite melt) and possibly metasomatized by $H_2O$- and/or $CO_2$-rich mantle-derived fluids [15] within zones of active rifting [81] may have deflected the liquid line of descent from the alkalic to the calc-alkalic trend without affecting the pristine mantle signature, thus giving rise to the Nb-rich calc-alkalic granite. Alternatively, the peralkaline affinity of the calc-alkalic granite may be explained by the early crystallization of Nb-rich minerals from an evolved mantle-derived peralkalic melt later affected by the progressive crustal melting, e.g. [82], but this model is not supported by the calcic composition of glass inclusions trapped in fergusonite.

## 6. Conclusions

Well-differentiated granite xenoliths occur in early-Pliocene alkali basalts in the northern part of the Carpathian back-arc basin (Pannonian Basin). Geochemical signatures and radiogenic isotopes are diagnostic of $A_1$-type within-plate granites inferred from fractionated, mantle-derived parental melts, whereas the calc-alkalic composition suggests crust-derived melts. Despite the calc-alkalic composition, the xenoliths exhibit strong peralkalic affinity, reflected in the inventory of primary magmatic HFSE-bearing minerals typical for silica-undersaturated alkalic granites and syenites: columbite-Mn, fergusonite-Y, oxycalciopyrochlore, Nb-rutile, and a Ca-niobate (fersmite/viggezite). The absence of allanite and monazite can be explained by low water and phosphorus contents in the parental magma. The calc-alkalic $A_1$-type granites intruding the upper crustal levels may represent an unconventional resource of critical metals.

**Supplementary Materials:** The following are available online at http://www.mdpi.com/2075-163X/9/8/488/s1, Table S1: Analytical conditions for electron probe micro-analysis (EPMA), Table S2: Representative EPMA and structural formulas of pyrochlore and samarskite, Table S3: Representative EPMA and structural formulas of fergusonite-Y, Table S4: Representative EPMA and structural formulas of columbite-Mn and Ca-niobate.

**Author Contributions:** Conceptualization, M.H.; methodology, software, and measurements, P.K.; original draft preparation, M.H. and V.H.; review and editing, V.H.

**Funding:** This research was funded by the VEGA grant agency, grant number 1/0143/18.

**Acknowledgments:** The original draft benefited from constructive remarks of two anonymous reviewers.

**Conflicts of Interest:** The authors declare no conflict of interest.

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
